# Adaptive tactile interaction transfer via digitally embroidered smart gloves

Yiyue Luo [1] ✉, Chao Liu[1], Young Joong Lee[1], Joseph DelPreto[1], Kui Wu [2], Michael Foshey [1], Daniela Rus [1], Tomás Palacios[1], Yunzhu Li[3], Antonio Torralba[1] & Wojciech Matusik [1] ✉

Human-machine interfaces for capturing, conveying, and sharing tactile information across time and space hold immense potential for healthcare, augmented and virtual reality, human-robot collaboration, and skill development. To realize this potential, such interfaces should be wearable, unobtrusive, and scalable regarding both resolution and body coverage. Taking a step towards this vision, we present a textile-based wearable human-machine interface with integrated tactile sensors and vibrotactile haptic actuators that are digitally designed and rapidly fabricated. We leverage a digital embroidery machine to seamlessly embed piezoresistive force sensors and arrays of vibrotactile actuators into textiles in a customizable, scalable, and modular manner. We use this process to create gloves that can record, reproduce, and transfer tactile interactions. User studies investigate how people perceive the sensations reproduced by our gloves with integrated vibrotactile haptic actuators. To improve the effectiveness of tactile interaction transfer, we develop a machine-learning pipeline that adaptively models how each individual user reacts to haptic sensations and then optimizes haptic feedback parameters. Our interface showcases adaptive tactile interaction transfer through the implementation of three end-to-end systems: alleviating tactile occlusion, guiding people to perform physical skills, and enabling responsive robot teleoperation.

Humans rely on their senses, including sight, hearing, and touch, to gather information about the environment and each other for everyday activities[1,2]. Physical tactile feedback, in particular, serves a critical role in learning, movement, communication, and environmental awareness[3–5]. As technology continues to evolve and its capacity to understand and assist more complex scenarios increases, there is a growing need to leverage such physical tactile experiences to enrich technology-mediated interactions among humans and between humans and machines[6,7]. Indeed, studies have demonstrated how sharing tactile information across humans and machines can be fundamental to personalized medicine and treatment[8,9], robot-assisted

surgery[10], human-robot interactions[11–13], augmented reality (AR)/virtual reality (VR)[14], and even everyday human activities[15,16].

Enabling intuitive tactile interaction transfer remains challenging since it requires scalable and conformal tactile sensing and haptic display systems that can be seamlessly integrated into our daily lives[17,18]. Nevertheless, there have been recent developments that aim to explore novel materials and fabrication methods that work to address such challenges. Developments of high-density, low-cost, conformal tactile sensing systems[19–22] have enabled the real-time capturing of physical tactile interactions during daily human activities, e.g., grasping and exercises, in a seamless manner. Also, to enable

[1]Department of Electrical Engineering and Computer Science, Massachusetts Institute of Technology, 32 Vassar Street, Cambridge, MA 02139, USA. [2]LightSpeed Studios, 12777 W Jefferson Boulevard, Los Angeles, CA 90066, USA. [3]Department of Computer Science, University of Illinois Urbana-Champaign, 201 North Goodwin Avenue, Urbana, IL 61801, USA. ✉e-mail: yiyueluo@mit.edu; wojciech@mit.edu

haptic feedback, a variety of techniques have been explored, including electrical[23–25], pneumatic[26,27], piezoelectric[28], ultrasonic[29–31], and electromagnetic systems[32–34], for human motor skill learning[35–37], immersive VR/AR interactions[25,38], and autonomous assistance[39].

To combine tactile sensing and haptic feedback, recent soft systems have explored hand-based epidermal closed-loop human-machine interfaces[40–43]. While previous works demonstrated the exciting potential of such systems, they also highlighted challenges such as their complex and delicate fabrication process that can limit scalability, robustness, customizability, and compatibility. Moreover, the variation in humans' perception of haptic feedback makes effective and reliable human-machine communication difficult since per-user calibration should also be minimized to create a seamless experience. Thus, scalable, compact, conformal, and adaptive human-machine interfaces with both tactile sensing and haptic display capabilities remain limited but obtain great potential for diverse real-world applications.

We aim to move towards addressing these challenges by presenting a textile-based wearable human-machine interface as well as its digital fabrication approach and adaptive control algorithms. We integrate tactile sensing and vibrotactile haptic feedback using a customizable and scalable computational fabrication pipeline, develop designs that can be applied to various domains, and create a machine-learning pipeline for per-user haptic optimization. Experiments demonstrate its ability to record, reproduce, and adaptively transfer physical interactions in a variety of contexts.

Focusing on tactile interactions of the hands, we present a diverse set of gloves with integrated tactile sensors and vibrotactile haptic actuators (Fig. 1a). Both the sensing and haptic components are automatically integrated into textiles with customized spatial resolution and positions using a digital embroidery machine. Each customized glove is fabricated within 10 min using low-cost commercial materials while maintaining a soft, conformal, and flexible nature of textiles. The vibrotactile matrices offer a spatial resolution of up to 4 cm², and the tactile sensing arrays achieve a spatial resolution of 0.25 cm². We investigate the usability and effectiveness of our interface through a user study, where 10 subjects evaluate and identify haptic feedback with different amplitudes, frequencies, temporal patterns, and locations on the hand.

We also develop a learning-based optimization pipeline to compensate for variations in users' perceptions by adaptively modeling individuals' responses to haptic feedback, which eliminates the need for manual calibration. The current implementation focuses on a paradigm in which the user should press a finger on the table when they feel haptic stimulation on that finger; this is used to guide a person through performing a tactile skill such as playing an instrument or playing a game. To initialize the model, we capture the responses of 12 users to the vibrotactile haptic feedback; we use this data to learn a forward dynamics model, which serves as a parameterized simulator for how a human might respond to the stimulation. Then to adapt this model to a new user, we introduce a module for few-shot adaption with a small amount of data; this allows the pipeline to optimize the haptic feedback for each subject over time while they are interacting with the system, improving the transfer of tactile information without a dedicated calibration routine.

Experiments demonstrate several applications of this textile-based wearable human-machine interface and its underlying algorithms. First, we aim to alleviate tactile occlusion by transferring forces sensed on the outside of the glove to haptic sensations applied on the person's hand inside the glove. Next, we demonstrate how transferring force sensations from a robot to a person can enrich teleoperation and enable more delicate grasping operations (Fig. 1c). Finally, we explore a skill-developing scenario that leverages the learning and optimization pipeline to adaptively transfer

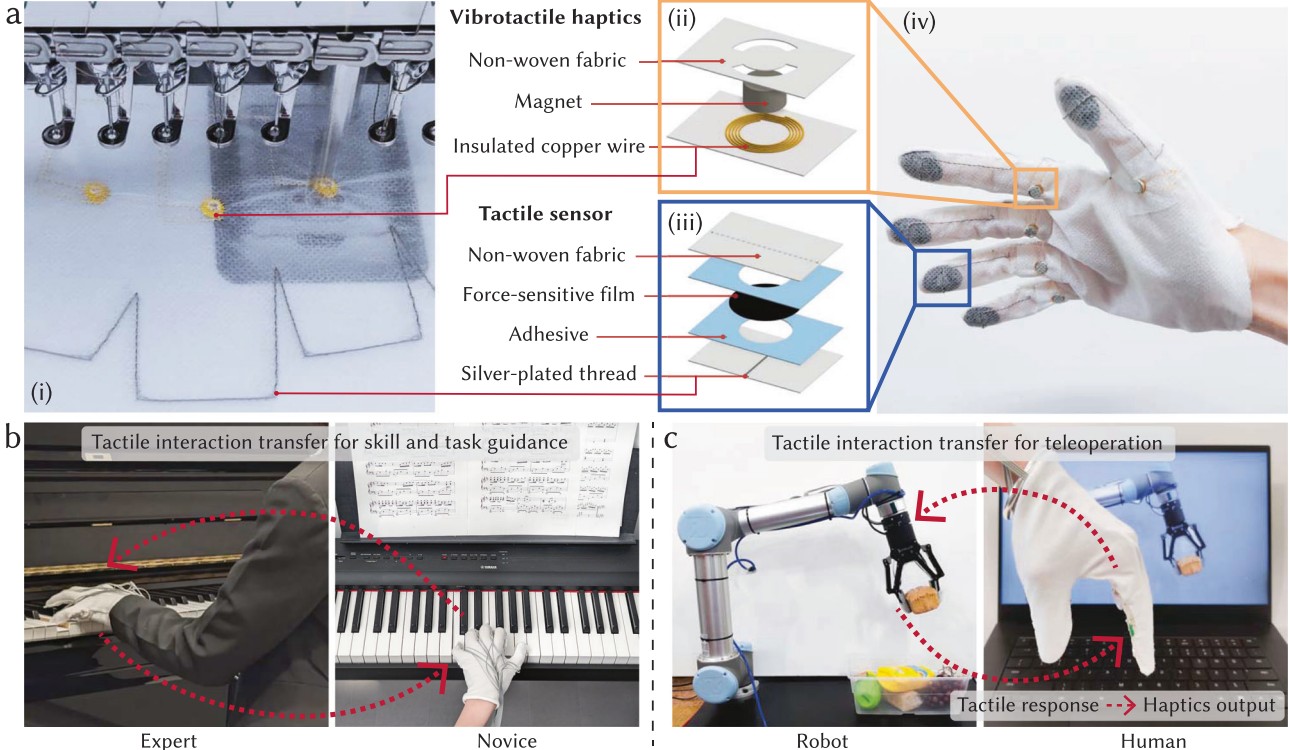

**Fig. 1 | Fabrication, structure, and applications of tactile interaction transferring smart gloves. a** A full-sized textile-based glove (iv) with integrated tactile sensors (ii) and vibrotactile haptics (iii) is digitally designed and automatically fabricated using a digital embroidery machine (i). Leveraging these gloves with integrated sensing and haptic capabilities, we demonstrate physical interaction transfer across people for skill development (**b**), and physical interaction transfer between humans and robots for teleoperation (**c**).

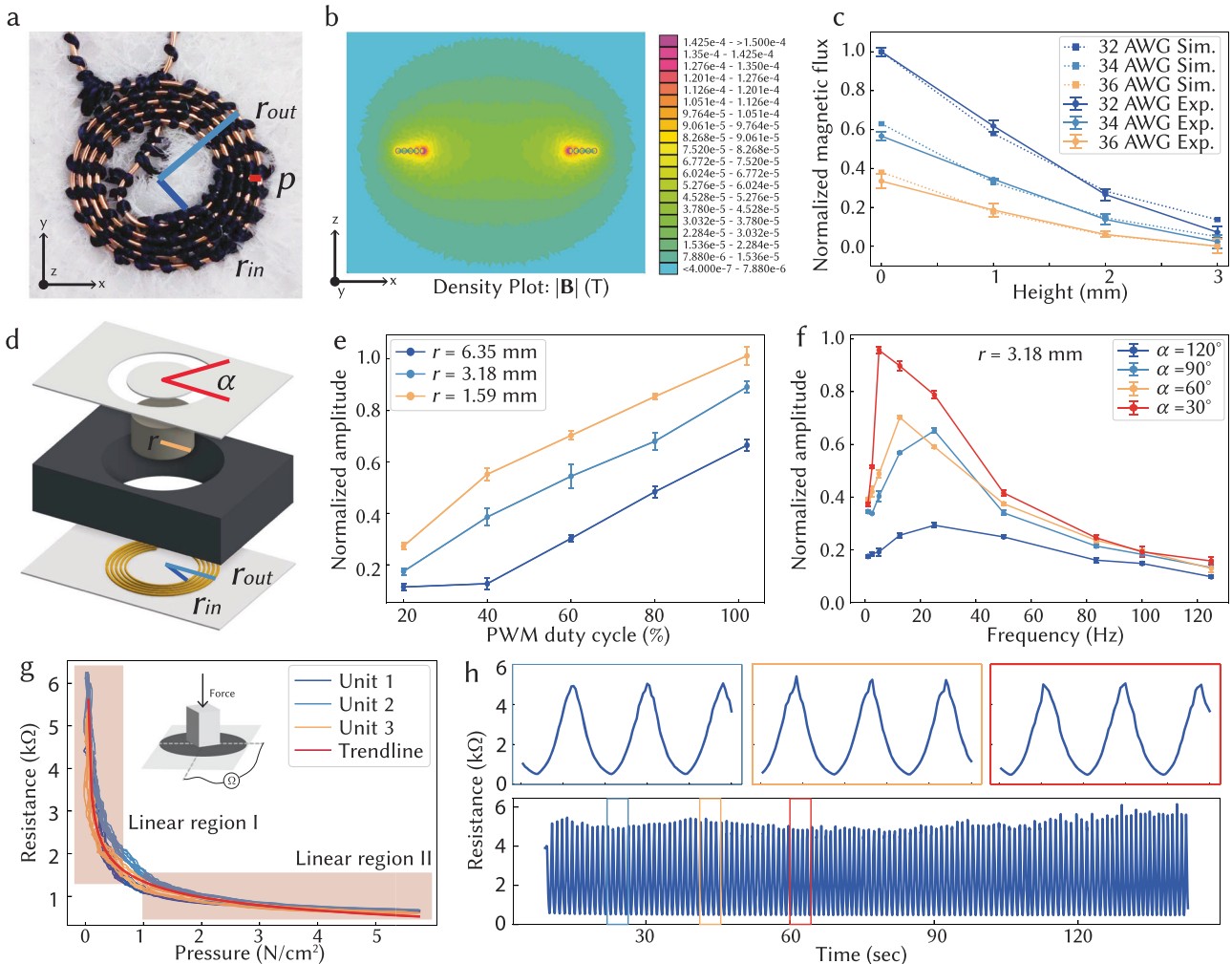

**Fig. 2 | Performance of embroidered vibrotactile haptics and resistive tactile sensors. a** Photograph on a typical embroidered magnetic coil showcasing inner and outer radii ($r_{in}$ and $r_{out}$) and the pitch ($p$) of neighboring coil winds. **b** Simulation demonstrating magnetic flux at the cross-section when subjected to a 1 A input current. **c** Comparison of normalized simulated and experimentally measured magnetic flux from embroidered coils with varying gauge numbers of magnetic wire at different heights. Error bars indicate the standard deviations (SD) across measurements. **d** Illustration of the characterization setup for a single vibrotactile unit with radius $r$, with a foam spacer providing base height anchoring and a single fabric link with angle $\alpha$ magnifying the vibrational displacement for analysis. **e** Vibration amplitudes of the vibrotactile unit increase with its sizes and the input PWM duty cycle. **f** The resonant frequencies of a typical vibrotactile haptic unit ($r = 3.18$ mm) decrease with smaller different fabric link angles. **g** Resistance profiles are consistent across three individual embroidered tactile sensors under pressure, with two linear regions. **h** Consistent performance of an embroidered tactile sensor was captured during 500 pressure (5 N/cm²) loading and unloading cycles.

tactile interactions from experts to novices and improve task performance (Fig. 1b).

In this way, we move towards enabling physical tactile interactions to persist across space and time and to be accessible simultaneously to multiple users and intelligent agents.

## Results

### Integrated tactile sensing and vibrational haptic feedback

The textile-based smart gloves, integrated with tactile sensing and vibrotactile haptic capabilities, are rapidly fabricated via a digital embroidery machine with minimal manual assembly (Fig. 1a and Supplementary Movie 1). The smart glove offers a comparable level of wearability to regular gloves, comparable in terms of size, flexibility, and weight. The structures of tactile sensors and vibrotactile units are demonstrated in Fig. 1a(ii, iii), respectively. The digital design and fabrication pipeline allows the smart gloves to be modularly designed and customized for the unique requirements of individual users or specific tasks; this includes rapidly adjusting the density, layout, and

size of tactile sensors or vibrotactile haptic units (Supplementary Figs. 1b and 2).

Our vibrotactile haptics are based on a linear resonant actuator structure, including an electromagnetic coil and a moving mass. More specifically, our typical vibrotactile haptic unit consists of two fabric layers, a bottom layer embroidered with enameled copper coils (Fig. 2a, 32 AWG, 5 winds), and a top layer with pre-cut symmetry circular slits. These slits generate a pair of symmetric fabric links with an angle of 30° accommodating a permanent magnet (K&J, N52 NdFeB) at the center. The two layers are affixed with thin adhesive. When an alternating pulse-width modulation (PWM) square wave signal is applied to the embroidered coil on the bottom layer (Fig. 2d), alternating magnetic flux is generated and leads to the vibration of the mounted permanent magnet on the top layer with a perceptible force and displacement. The symmetric pre-cut slits on the top layer allow for vertical movement of the attached magnets while effectively securing them at specific locations. The vibration frequency and amplitude of the haptic unit are contingent upon various factors,

including the design of the embroidered copper coil, the size of the permanent magnets, and the duty cycle of the alternating PWM input. The embroidered enameled copper traces obtain a tensile strength of 20 N and endure 15% elongation and thousands of bending cycles (Supplementary Fig. 3).

Figure 2a, b demonstrate a representative design of an embroidered copper coil and the corresponding simulated magnetic flux heatmap (with 1 A alternating current input), respectively. The measured magnetic flux from the embroidered magnetic coils (subjected to a full-scale duty cycle input) agrees with the simulation result (Fig. 2c). The magnetic flux generally decreases with the increase in sensing height and magnetic wire gauge. Furthermore, the displacement of the vibrotactile haptic units can be captured and quantified through a high-speed camera (Supplementary Movies 2, 3 and Supplementary Fig. 4c). To enhance the haptic actuator response for quantitative analysis, a foam spacer was added to anchor the base height level and only one of the symmetric fabric links was obtained to amplify the displacement. The vibrotactile displacement increases with the rise of the input PWM duty cycle and with the decrease of permanent magnet size, which obtains higher magnetic field-to-weight ratios (Fig. 2e). Figure 2f presents the normalized vibration amplitudes of a typical vibrotactile haptic unit (with a radius of $r = 3.18$ mm) across various left-out fabric link angles ($\alpha$) and alternating input frequencies, effectively highlighting the resonant frequencies of different designs. Notably, as the fabric link angle decreases, the vibration amplitude intensifies, while the resonant frequency concurrently decreases.

Our typical tactile sensor comprises a piezoresistive layer placed in between two fabrics with embroidered conductive silver yarn electrodes (Fig. 1c). It obtains a minimum detection limit of $0.35$ N/cm$^2$ and a maximal detection limit at 20 N/cm$^2$ (Supplementary Fig. 5a). When the applied normal pressure is increased up to 6 N/cm$^2$, the resistance of tactile sensors drops from 6 to $0.8$ k$\Omega$, with two linear sensing regions Consistent performance was retrieved from multiple individual tactile sensors, reiterating the advantages of standardized digital design and fabrication pipeline (Fig. 2g). Our sensor functions with minimal hysteresis during pressure loading and unloading cycles at with minimal hysteresis (Supplementary Fig. 5b). The stability and durability were further demonstrated by the stable sensing performance of our sensors under 2000 pressure loading and unloading cycles (Fig. 2h and Supplementary Fig. 5c).

Both tactile sensors and vibrotactile haptic units are serialized in a matrix-based form factor to minimize the number of connecting wires (Supplementary Fig. 6a, b). The tactile sensors are connected to a modified electrical-grounding-based circuit architecture (Supplementary Fig. 6a) to eliminate most cross-talk and parasitic effects of the passive matrix[44]. We arrange the vibrotactile haptic units in a matrix, where each row and each column is connected to a half H-bridge[45] composed of two N-type MOSFETs in series, as shown in Supplementary Fig. 6b. We turn on the half H-bridge along a specific row and column to activate a specified vibrotactile unit in a multiplexed manner. The intensity and frequency of the vibrations are fully programmable. The direction of displacement and the polarity of generated magnetic flux is determined by the direction of current flowing through the coil, which is controlled by the switches of the H-bridge. The intensity of the vibration relies on the average voltage and current delivered to the coil, which is controlled via PWM. The vibration displacement of three neighboring vibrotactile units shows a consistent displacement with minimal interference between each unit when activated sequentially and alternatively (Supplementary Fig. 6c, d).

## Human perception of vibrotactile haptic feedback
We first evaluate the effectiveness of our platform by exploring how people perceive the textile-based vibrotactile haptic feedback. We conducted a user study with 10 subjects (aged 26–32 years, 4 females) that investigated their perceptions of haptic feedback with different amplitudes and frequencies. Figure 3b, c show their resulting ratings of perceived amplitude. In general, users perceive the feedback more strongly with higher input amplitudes and at a frequency of 100 Hz.

In addition, tests were performed to explore whether users could identify where haptic feedback is being applied on the hand, and whether they can distinguish between temporal patterns of activation. Wearing a glove with 23 embroidered haptic units distributed across the inner hand as shown in Fig. 3a, g), participants were able to identify the activated vibrotactile unit (full PWM duty cycle at 100 Hz) with an average accuracy of 94% (Fig. 3d). Users also identified temporal patterns of activating vibrotactile units with an average accuracy of 92% (Fig. 3e). Notably, the discriminative capability remains unaffected by interference from a constantly vibrating haptic unit (Supplementary Fig. 7). These results are promising for effectively conveying spatial and temporal information and for using the textile-based system integrated with tactile sensing and vibrotactile haptic capabilities to transfer physical tactile interactions.

## Tactile interaction transfer for a single user
The sense of touch is essential for delicate manipulation tasks, but it can be significantly hindered when occluded by gloves. This is particularly evident in situations where astronauts and technicians must wear thick personal protection equipment for dangerous tasks, resulting in a considerable reduction in tactile perception. Such tactile occlusion could be alleviated by transferring the physical sense of touch from the glove's outer contact area to the human skin on the glove's inner surface. This would allow users to regain their perception of the external environment, which is crucial for dexterous manipulation tasks. Leveraging the textile-based smart gloves with integrated tactile sensing and vibrotactile haptic capabilities, we demonstrate the transfer of coarse pressure-based tactile interactions to alleviate tactile occlusion.

We develop a double-glove interface by augmenting a hand-shaped piece of fabric with 23 tactile sensors, then attaching this to a thick protective glove for animal handling (Fig. 3f). We then fabricate a glove with 23 vibrotactile units whose positions correspond to the sensor locations (Fig. 3g), and insert this actuated glove into the sensorized protective glove. The completed system can capture tactile interactions outside the protective glove and transmit it to the human skin via the inner haptic glove in real-time (Supplementary Movie 4).

To evaluate this system, we asked users to identify the location of a contact force when the haptic actuators were activated or deactivated. As illustrated in Fig. 3h, users achieved an accuracy of 88.6 ± 2.62% with haptic feedback activated and an accuracy of 5.7 ± 4.4% without the haptic feedback. This proof of concept explored a coarse form of tactile transfer, where results are promising for using the textile-based system with integrated tactile sensing and vibrotactile haptic capabilities to help alleviate tactile occlusion.

## Adaptive tactile interaction transfer across users
Leveraging our textile-based wearables, we are able to capture the real-time tactile interactions from an individual user and then reproduce and transfer such interactions to another user via haptic instructions. For example, we ask a piano teacher to play a certain rhythm and capture the corresponding tactile sequence ($T_{target}$). We then derive the haptic instruction sequence ($H$) based on the captured data and display it to a student through a haptic glove. Based on the haptic instruction, the student is expected to reproduce the rhythmic sequence of the teacher (Supplementary Fig. 8 and Supplementary Movie 5).

We measure the difference between the students' reproductions and the target tactile sequence to evaluate the transfer efficacy. We assume two users achieve the same performance for a given task when they produce the same tactile signals during the interactions; a smaller difference between a student's reproduced tactile sequence ($T$) and the teacher's ($T_{target}$) indicates a more effective transfer.

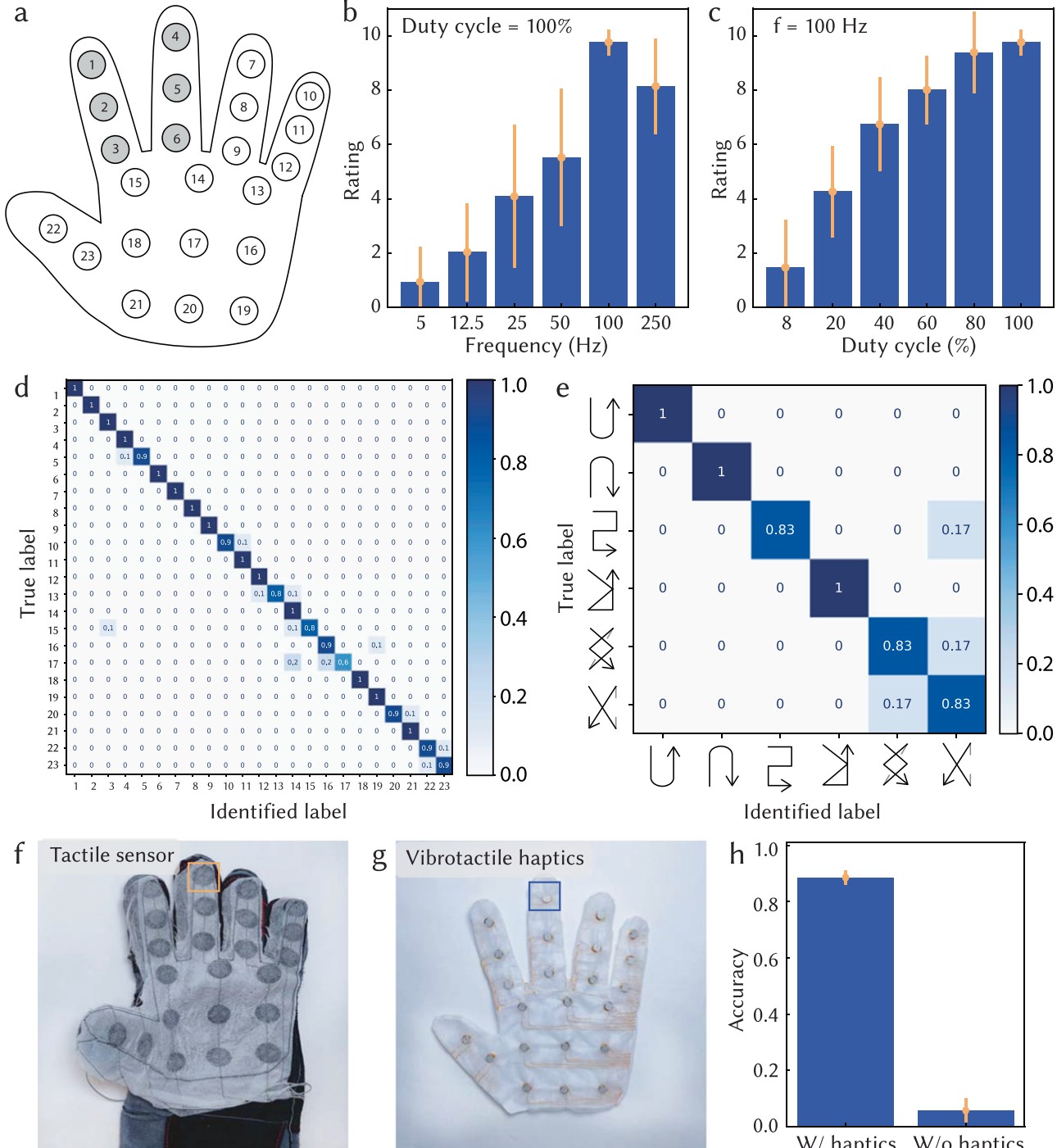

**Fig. 3 | Users' perception of vibrotactile haptic feedback. a** Illustration show-casing the placement of vibrotactile units on a smart glove for studying users' perception. **b, c** Normalized user ratings on a scale of 0 to 10, indicating the perceived strength of haptic feedback for various amplitudes and frequencies. A rating of 0 represents no perception, while a rating of 10 signifies the strongest perception. **d, e** Confusion matrices summarize users' identification of which unit is activated and temporal activating patterns (230 trials for position identification and 60 trials for temporal sequence identification). The temporal activating patterns are demonstrated as the confusion matrix labels, which happen at the vibrotactile units on the thumb and index finger, colored in gray. Photograph of an outer sensing glove (**f**) to capture tactile cues and an inner haptic glove (**g**) to transfer tactile cues under tactile occlusion. **h** Accuracy of classification of single contact points under tactile occlusion, with and without the haptic feedback. Error bars indicate the standard deviations (SD) across measurements.

A challenge is that simply producing an $H$ that directly copies or linearly maps $T_{target}$ will not result in a faithful reproduction since each person reacts differently to the same haptic sequence. Moreover, a single person may also react differently to haptic signals at different hand locations and with different timings. Therefore, to ensure the effectiveness of tactile interaction transfer, the haptic instructions must be adaptively optimized across time, subjects, and task requirements.

To address this challenge, we develop an adaptive human model learning and inverse haptics optimization pipeline to derive personalized vibrotactile haptic instructions[46]. It outputs an optimized haptic signal ($H$) given an input target tactile sequence ($T_{target}$). The pipeline

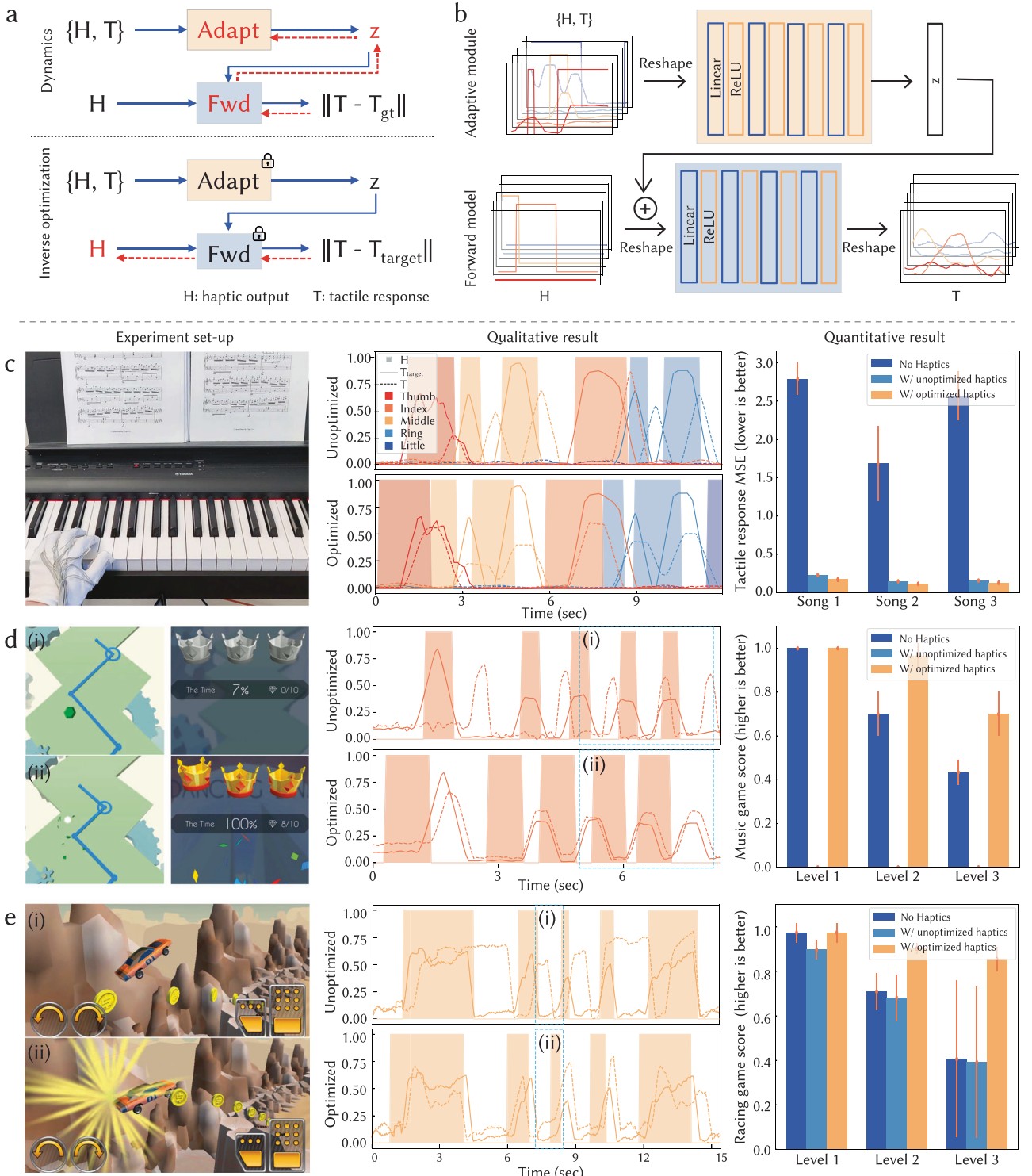

**Fig. 4 | Adaptive transfer of tactile interactions across users. a** An overview of human model learning and inverse haptics optimization pipeline. The pipeline incorporates a forward dynamics model with an adaptive module and an inverse optimization procedure. The vibrotactile haptic output is denoted as *H*, and the tactile response as *T*. Backward propagation is represented by the red dashed arrows. **b** Illustration of the forward model featuring an adaptive module. Using our pipeline, we demonstrate improved performance in piano playing (**c**), rhythmic gaming (**d**), and racing gaming (**e**) by transferring tactile interactions from expert users. The experimental images (left column) were captured during the time intervals indicated by the dashed light blue boxes depicted in the qualitative results plots (middle column). Error bars indicate the standard deviations (SD) across measurements.

comprises a forward dynamics model and an inverse optimization process, as illustrated in Fig. 4a. The forward dynamics model simulates a user's reaction to the haptic instruction by taking in a sequence of haptic signals (*H*) and predicting the resultant tactile sequence (*T*).

Given a trained forward dynamics model, the inverse optimization process aims to minimize the difference between the predicted tactile sequence (*T*) and the target tactile sequence ($T_{target}$) by optimizing the parameters of the haptic sequence *H* using gradient descent.

To adapt the pipeline to individual users without dedicated calibration routines, we include an adaptive module that takes in a combined series of haptic instructions and tactile responses ($\{H, T\}$) and generates a user-specific latent feature vector ($z$). This latent encoding may represent information such as reaction time and finger dexterity (Fig. 4b). The feature vector ($z$) is then provided to the forward dynamics model together with the haptic sequence. By including the adaptive module, our pipeline learns to adapt to the individual user's response to the haptic instructions and optimizes the haptic sequence output for individuals given a specific tactile sequence goal. This enables an adaptive, automatic, and accurate conversion between tactile and haptic signals. This process can occur dynamically, allowing the interface to improve in real-time.

Using a smart glove with sensing areas at the fingertips and haptic units on the inner proximal phalanges, we collected a 12-person dataset of tactile responses to pre-generated haptic instructions. As a baseline, we trained a forward model for each subject individually. Then we trained a universal forward model that omits a selected subject (Supplementary Data 1) and evaluated its performance both with and without an adaptive module. As shown in Supplementary Fig. 9a, using an adaptive module reduced the mean-squared error (MSE) and even achieved comparable performance to the per-subject baseline model. Compared to baseline models trained for individual users, our adaptive model allows generalization and adaptation to new users with 15 s of user data.

It can be seen that the predicted tactile response from the trained forward model aligns with the ground-truth tactile signal (Supplementary Fig. 9d). The effectiveness of the pipeline is also validated by comparing a user's tactile response with the given target tactile sequence using both the unoptimized and optimized haptic instructions (Supplementary Fig. 9e). As suggested by Supplementary Fig. 9f, the optimized haptic instructions are able to compensate for the user's reaction time to haptic instructions.

We apply this learning and optimization pipeline to piano playing, rhythmic gaming, and racing games (Fig. 4c–e; Supplementary Fig. 10 and Movie 6). In the piano-playing scenario, users are asked to play a specific sequence that can be achieved without moving their hand to new keyboard locations. Performance is evaluated by comparing the tactile response from the users to the target tactile response captured by a piano expert. In the rhythmic game, users are expected to follow a designed track and bump into the goal area (green dot) by left-clicking the mouse to switch the moving line (blue line) direction. Users need to stay on the route (light green area) and are scored based on how many goals they bump into. In the racing game, users are asked to reach the finish and collect coins along the way by controlling the balance of the car. Performance is evaluated with the number of collected coins. In all scenarios, we first record the tactile response sequence from a high-performance player's finger pressing; taking the recorded tactile sequence as input, the optimization pipeline outputs the optimized haptic instruction for a specified user. We record and compare the performance of that specific user with optimized haptic instructions, unoptimized haptic instructions, and a baseline that does not provide any haptic instructions.

The qualitative results shown in Fig. 4c–e demonstrate that given the adaptively optimized haptic instructions, users are able to reproduce a similar tactile interaction with aligned timing based on the target tactile interaction sequence. In general, for simpler levels, users tend to have comparable performance regardless of optimized haptic guidance. However, as tasks get harder, users perform better when the optimized haptic guidance is provided. In time-sensitive scenarios, e.g., the rhythmic game, we observe that unoptimized haptic instruction confuses users and thus lowers users' performance. Additionally, users tend to fail levels more easily with unoptimized haptic instructions since their errors accumulate over time. Adaptive optimization of haptic instructions, on the other hand, improves users' performance by guiding their actions at the right time during harder levels.

## Tactile interaction transfer for teleoperation

Sharing physical tactile interactions between humans and robots plays a critical role in intuitive human-robot collaboration and interactive teleoperation. This is especially important when visual information is occluded, which often happens in real-world scenarios. We demonstrate that our textile-based smart gloves integrated with tactile sensing and vibrotactile haptic capabilities enhance teleoperation performance for fragile and soft object grasping by transferring the physical interaction captured from a parallel robotic gripper to the user's hand in real-time. We equip a parallel robotic gripper (Robotiq, UR5 arm) with a textile-based tactile sensing array (3 sensing areas on each side). Then, we ask the user to wear a designed haptic glove with integrated vibrotactile units at the corresponding locations along the thumb and index finger. We embroider colored markers on the haptic glove, which enable the tracking of the distance between the thumb and index finger with a camera in real-time. This estimated distance is used to control the separation of the robot's parallel gripper (Fig. 5a).

To evaluate our approach, we tele-grasp a hot dog bun, a burger bun, Hawaiian bread, and a seaweed plastic box. When visual feedback is available, the user is generally able to perform relatively optimal grasps, where the soft objects only deform slightly to an average of 87% of the natural object width regardless of whether haptic feedback is provided (Supplementary Fig. 11a). In this process, the user infers the grasp quality from the visual deformations of the objects. On the other hand, when visual information is obscured, we observe that haptic feedback significantly improves the tele-grasping quality.

In the absence of visual and haptic feedback, the user has no information about the grasp and tends to apply strong grasps, resulting in the objects being squeezed to an average of 58% of the natural object width. In contrast, when real-time haptic feedback is provided, the user is able to perceive whether the grasp is good enough through the vibrotactile feedback, even without visual indications of which object is being grasped. This allows the user to control the gripper for a more optimal grasp, securing the object with relatively minimal deformation to an average of 95% of the natural object width (Fig. 5b, c and Supplementary Movie 7).

## Discussion

Our work focuses on developing a scalable, customizable, wearable textile-based human-machine interface for capturing, displaying, and adaptively transferring physical tactile interactions. The tactile sensing array and vibrotactile haptic array are digitally designed and fabricated, and seamlessly integrated into smart gloves. Experiments demonstrate applications for reproducing coarse tactile interactions between humans and the environment, allowing people to share tactile sequences, and enriching robotic teleoperation.

The integration of sensing and haptic actuation happens within the same fabrication cycle in a scalable and customizable manner. We are able to construct smart gloves integrated with tactile sensing and vibrotactile haptic capabilities in a modular way to accommodate different users or task specifications. Furthermore, we present a learning-based approach to adaptively optimize haptic output for individual users, without relying on manual calibration periods. We demonstrate a proof of concept of this behavior modeling and adaptive control approach for guiding users to perform touch sequences. In piano playing, rhythmic gaming, and racing gaming scenarios, we demonstrate that our interface and optimization pipeline are effective for adaptive tactile interaction transfer across users.

Given that humans are most sensitive to haptic feedback on their hands, our system only focuses on tactile interactions on the hands so far. However, with the scalable and customizable digital design and fabrication pipeline, we can extend the system to other wearables,

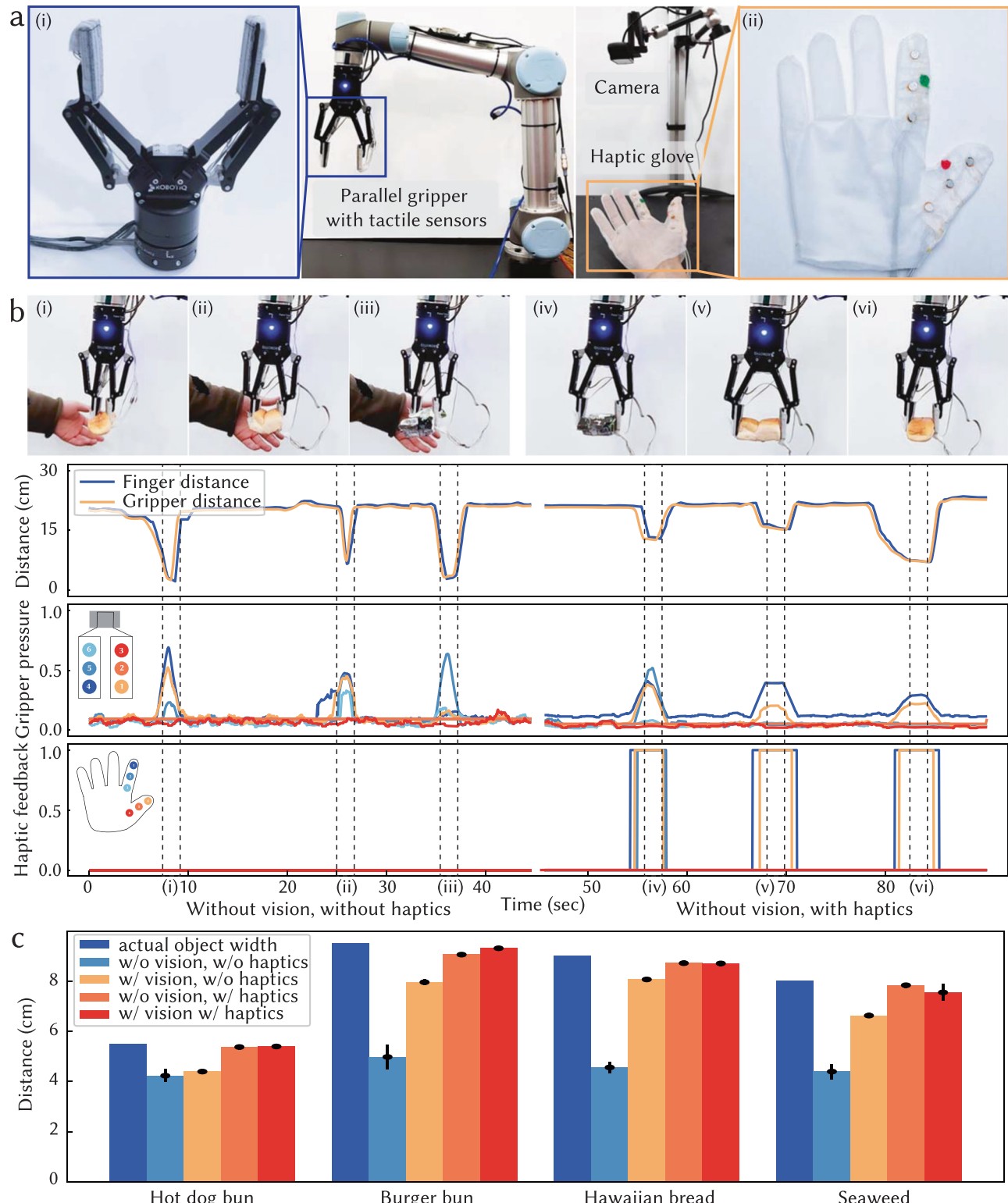

**Fig. 5 | Transfer of tactile information from robot to human for teleoperation.**
**a** Overview of the teleoperation setup, featuring a parallel gripper equipped with
tactile sensors (i) and a user wearing a full-sized glove integrated with vibrotactile
units on the thumb and index finger (ii). The visual system tracks embroidered
green and red color blocks for distance measurement purposes. **b** Teleoperated
grasping of a hot dog bun, Hawaiian bread, and a soft plastic box without visual
feedback and with optional tactile information transferring through haptic feed-
back. The plots show the synchronized distance measurement of users' finger
manipulation and parallel gripper control, normalized pressure captured from the
parallel gripper, and mapped haptic feedback to the user. With haptic feedback, the
user tended to teleoperate the gripper such that objects were securely grasped with
reduced pressure and smaller deformations. **c** Quantitative measurements of the
actual width of objects and the distance between grippers during tele-grasping
under different scenarios. Gripper distances, which are smaller than the actual
object width, imply deformation of the object. Error bars indicate the standard
deviations (SD) across measurements.

such as socks and jackets, to provide haptic feedback to other body parts.

It is also worth noting that the movement of hands can affect users' haptic perceptions, and metal objects may interfere with the vibrotactile haptic units. These limitations can be addressed by fabricating more conformal and tight wearables with additional insulating layers. Currently, our adaptive optimization pipeline does not consider the AC input amplitudes and frequencies because users' perception of vibration amplitude varies significantly over time; it instead focuses on the users' reaction times. Nevertheless, by collecting more data on users' behavior, the pipelines can be expanded to include parameters on the amplitudes, frequencies, and temporal patterns of vibration, resulting in a larger design space.

In general, our textile-based human-machine interface offers a scalable and adaptive approach for exchanging and transmitting physical tactile interactions across people and time. We demonstrate that such tactile interaction transfer alleviates tactile occlusion, promotes skill and task performance, and facilitates robot teleoperation. The potential applications of our work are wide-ranging and extend beyond the lab, including use in training programs for various skilled professions such as surgeons, pilots, and engineers. This textile-based wearable human-machine interface could also be integrated into virtual reality and augmented reality systems, providing users with a more immersive and tactile experience. Furthermore, the interface's ability to adaptively transfer tactile information between humans and machines could be used in industries such as manufacturing, where robots and humans can work collaboratively with increased safety and efficiency. We envision that our system will enable physical tactile interactions to persist across space and time and be accessible simultaneously to multiple users.

## Methods

### Digital machine embroidery
Embroidery is one of the most common textile processing methods for garment fabrication and decoration[47]. Machine embroidery has been well-established and widely used in the textile industry for mass production[48]. Machine embroidery uses two threads, an upper thread and a bobbin thread, which run along the top and bottom of the fabric to generate interlaced locked stitches. We leveraged a computational design pipeline, which takes in a scalable vector graphic file (SVG) and a specified stitch pitch, and outputs a drawing sheet set (DST) encoding a full list of digital embroidery machine operations stitch by stitch (Supplementary Figs. 12 and 13). The digital embroidery machine instruction was then sent to an industrial-scale embroidery machine (Tajima TWMX-C1501, Supplementary Fig. 14) for fully automatic fabrication. Thanks to the computational design and automatic fabrication pipeline, the design of textile-based smart gloves can be fully customized in terms of sensors or haptic units' size, positions, density, and so on.

### Fabrication of vibrotactile haptic units
The magnetic wire (BNTECHGO 32 AWG enameled copper wire, Supplementary Fig. 15) was embroidered as an electromagnetic coil on the fabric substrate at the speed of 700 stitches/min. Since the copper thread is stiffer than the usual embroidery threads (Madeira 100% polyester), we used it as a bobbin (bottom) thread, which experienced less strain during the embroidering process and thus ensured a stable fabrication process. Different design parameters of the embroidered copper coil, including inner radius $r_{in}$, outer radius $r_{out}$, coil pitch $p_c$, and stitch pitch $p_s$ are investigated and optimized (Supplementary Fig. 1b). Based on the size of magnets (diameter = 6.3 mm) and the size of copper thread (diameter = 0.2 mm), we set the inner radius $r_{in}$ as 3 mm, outer radius $r_{out}$ as 4 mm, coil pitch $p_c$ as 0.25 mm (Fig. 2d). Stitch pitch $p_s$ was set as 1 mm to maintain the optimal tension of the embroidered coil. The top fabric was first laser-cut to generate a

circular slit with pre-defined angles. Commercial permanent magnet (K&J, N52 NdFeB, thickness = 1.6 mm) of different sizes (diameter = 3.2 mm, 6.3 mm, 12.7 mm,) was affixed to the center of slits by thin adhesive (3M 468MP). The top fabric with the affixed magnet was aligned to the embroidered coil matrix and affixed with adhesive. In this work, the vibrotactile haptic units were fabricated using nonwoven fabric as a substrate; however, the same fabrication process is feasible for a wide range of substrates (Supplementary Fig. 1a)

### Fabrication of tactile sensing units
Each of the tactile sensing units is based on a triple-layer design, where a piezoresistive layer (Velostat, Supplementary Fig. 15) was sandwiched between two fabric substrates with embroidered conductive yarns (Madeira HC40, polyamide/silver plated) as electrodes. We first embroidered the horizontal and vertical electrodes on separate fabrics. We then sandwiched the piezoresistive film at specific locations between the two fabrics using shaped adhesive (3M 468MP, Supplementary Fig. 15), which was pre-cut hollow in the center to enable direct contact between the electrodes and piezoresistive film (Supplementary Fig. 12b(i, ii)).

### Fabrication of smart gloves integrated with tactile sensors and vibrotactile haptics
The fabrication of full-size textile-based gloves with seamlessly integrated tactile sensors and vibrotactile haptic units is composed of 6 steps. First, we embroidered the silver-plated conductive thread as horizontal and vertical electrodes for the tactile sensors (Supplementary Fig. 12b(i)). We then affixed the piezoresistive layer at specific locations and flipped the fabric with vertical electrodes over to the one with horizontal electrodes, forming the three-layer sensing structure (Supplementary Fig. 12b(ii)). The electromagnetic coils were then embroidered onto the same fabric (Supplementary Fig. 12b(iii)). To further offer flexibility to the glove design, we applied Spandex fabric to the bottom of the embroidered fabric and stitched the outline of the glove design (Supplementary Fig. 12b(iv)). We then cut the excess fabric (Supplementary Fig. 12b(v)). Lastly, we flipped the embroidered glove inside out to avoid visible seams and aligned and attached the laser-cut top fabric with the tagged permanent magnets onto the embroidered coil matrix. The tactile-vibrotactile glove would be fully functional after connecting the tactile sensing electrodes and the extended copper thread to the readout and driving circuits (Supplementary Fig. 12b(vi)).

### Simulation of magnetic flux in vibrotactile haptic unit
We simulated the magnetic flux generated by the embroidered coil with different designs. In particular, we performed a finite element method (FEM) simulation to compute the magnetic flux density and compared it with the actual magnetic flux density we measured from the physical samples. All our simulations are conducted using Finite Element Method Magnetics (FEMM)[49]. We considered our design as a symmetric structure and used FEMM to solve the axisymmetric problem with one cross-section slice, as shown in Fig. 2b and Supplementary Fig. 4a, b.

### Characterization of vibrotactile haptic unit and tactile sensor
The magnetic flux of different embroidered coil designs was measured by a Gauss meter (FW Bell 9500) using a probe of around 5 mm by 5 mm surface area. The vibration amplitude of the haptic units was recorded by a 240 fps camera and extracted through ImageJ. The interference between two neighboring haptic units was investigated by activating each unit sequentially at 100% duty cycle (input voltage of 1.7 V) at 50 Hz. The tensile strength and maximum elongation of the fabrics with embroidered vibrotactile haptic units were evaluated by tensile test via a mechanical tester (Shimadzu AGS-X), as shown in Supplementary Fig. 3a. Numerous tensile testing cycles were

conducted to assess the durability of embroidered enameled copper wire (Supplementary Fig. 3c, d). The resistance profile of a typical tactile sensor was measured by a digital multimeter (Tektronix DMM4050) with a loading normal pressure of up to 35 N/cm$^2$ (Shimadzu AGS-X). A 2000-cycle test was performed on 3 individual tactile sensors by applying 0.35 to 5 N/cm$^2$ loading and unloading cycles at the rate of 3 mm/min.

### Tactile readout and haptic driving circuit

The tactile sensors were serialized through an electrical-grounding-based circuit architecture in a matrix form factor[44]. A reference voltage $V_{ref}$ of 2.5 V was applied to each column. An amplifier was added to each column with the gain resistor $R_g$ of 1 kΩ. A capacitor of 10 µF was added in parallel with each gain resistor to reduce noise. Controlled by Arduino Nano, each row (A2, B2, C2) was grounded at a time while each column (A1, B1, C1) was multiplexed through for signals from individual sensors. Measurements were transformed into a 10-bit digital signal and transmitted serially to a computer.

The vibrotactile haptic units were controlled with a customized circuit mounted on a microcontroller board. Our circuit consists of a matrix of five H-bridges, Schottky diodes, and connectors for interconnection between haptic units. Each haptic unit was driven by an H-bridge matrix, where each row and column was connected to a half-bridge circuit architecture, respectively. The half H-bridge circuit comprised two N-MOSFETs (IRLB8721PbF, Infineon Technologies) connected in series. Forming a full H-bridge using specific pairs of horizontal (e.g., A11 and A12) and vertical (e.g., A21 and A22) half H-bridges, the defined vibrotactile haptic units (e.g., top-left coil) can be activated in a multiplexed manner. All half H-bridges were controlled by a microcontroller board (Arduino Mega 2560 Rev3) through pulse-width modulation (PWM) pins (e.g., A11 and A21) and digital pins (e.g., A12 and A22). Coils were driven by square wave input, where the PWM duty cycle is similar to the amplitude of the input signal. Each coil was separated by a low turn-on voltage diode (Part number) to prevent cross-talks. External power was supplied by a portable lithium-ion polymer battery (3.7 V, 2500 mAh) to drive the coils.

### Experiments with human subjects

Our experiments with human subjects were approved by the Committee on the Use of Humans as Experimental Subjects (COUHES) at the Massachusetts Institute of Technology (MIT, 2210000769). Participants were recruited through verbal advertisements and voluntarily participated in the studies without receiving any compensation.

**Tactile perception with different input amplitudes and frequencies.** We investigated users' perception of vibrotactile feedback with different input AC amplitudes and frequencies. We recruited 10 human subjects, who were provided with haptic feedback under different input modulation (PWM duty cycle of 100%, 40%, 20%, 10%, 5% at a frequency of 100 Hz) amplitude and frequency ($f$ = 250, 100, 50, 25, 12.5, 5 Hz at 100% PWM duty cycle). Participants then provided quantitative feedback based on the amplitude of sensations. All participants were right-handed. In a randomized order, different AC inputs were continuously applied to participants' index fingertips on left hands until they provided corresponding quantitative ratings.

**Distinguishing spatial and temporal patterns of haptic feedback.** In this experiment, 10 recruited participants were asked to wear a vibrotactile haptic glove with 23 integrated vibrotactile units. We activated one random unit at a time and asked the participants to identify the location of their sensations (23 trials per participant). We investigated the interference effect by repeating the same experiment with one of the haptic units continuously activated (23 trials per participant, Supplementary Fig. 7). Furthermore, we activated 6 vibrotactile haptic units along the index and middle finger with some

defined temporary sequence and asked the participants to classify the spatial and temporal pattern (10 trials per participant). Each sequence is around 3 s. Classification accuracy is reported in Fig. 3d, e. All vibrotactile haptic units were activated with 100% PWM at 100 Hz.

**Transfer physical tactile interactions for a single user.** We asked 2 recruited participants to wear double gloves, which consist of an outer thick animal handling glove (RAPICCA) with mounted tactile sensors, and an inner regular-sized haptic glove. The tactile sensors capture the pressure information, which is converted to the corresponding vibrotactile haptic feedback (PWM duty cycle of 100% at 100 Hz) based on threshold clipping (the corresponding haptic units were activated if the tactile sensing reading exceeded 50% in weighted amplitudes) in real-time at the frame rate of 40 Hz (Supplementary Movie 4). We then applied a force stimulus to a specific area on the animal handling glove and asked participants to locate the stimulus with and without the vibrotactile feedback. Accuracy is reported in Fig. 3h.

**Transfer physical tactile interactions across users.** We demonstrated physical interaction transfer across people through the task of piano playing. We recruited 2 human subjects, one had more than 10-year piano-playing experience, acting as the teacher, and the other was a beginner in piano playing, acting as the student. Both participants wore a smart glove, which consisted of 5 tactile units on the fingertips and 5 haptic units on the inside of each finger (Supplementary Fig. 8a). We first investigated physical interaction transfer by asking the teacher to play a very simple music sequence and recording his/her finger pressing sequence, duration, and amplitude through the serialized tactile signal. We then converted the pre-recorded tactile signal to haptic feedback based on a linear mapping, according to which, the student was asked to perform sequence pressing. The student's performance was recorded by the tactile units at the fingertips, which was later quantified by comparing it with the pre-recorded tactile sequence (Supplementary Movie 5 and Supplementary Fig. 8b, d). This process can also happen in real-time, where the tactile signal from the teacher's playing sequence is retrieved and converted to haptic feedback on the student's glove simultaneously (Supplementary Movie 5 and Supplementary Fig. 8c, e).

### Human model learning and inverse haptics optimization

**Dataset.** Finger pressing is a common action for diverse daily activities, including playing piano, gaming, typing, and so on. Using finger pressing as a concept-proofing task, we collected data from 12 participants (IRB by MIT 2210000769). Participants wore our designed smart glove, where 5 tactile units were located at the fingertips and 5 haptic units were located at the inner side of each finger (Supplementary Fig. 8a). Participants were informed about the basic rule, which was to press the corresponding finger whenever they perceived vibration on one of the five fingers. We played randomly generated haptic instructions, where the 0-1 haptic unit was activated for a random duration. The vibrotactile haptic units were activating at full modulation at the frequency of 100 Hz. The sensing response from the participant's pressing was captured in real-time. More than 150,000 frames were collected (around 5 min of continuous pressing per participant). We split data from each individual participant as train, validation, and test sets with a ratio of around 6:2:2.

**Forward dynamics model.** The forward model took in a sequence of haptic sequence ($H$, 200 frames, around 4 s) and output a sequence of predicted tactile signals ($T$) with the same length. The model was implemented in PyTorch, composed of 4 fully connected layers, each of which was followed with a ReLU activation function (Fig. 4b). Mean-squared error (MSE) was computed and minimized between the predicted tactile signal ($T$) and ground-truth tactile signal ($T_{gt}$) from the participant's pressing sequence during data collection (Fig. 4a). The

adaptive module took in a concatenated sequence of haptic sequence and tactile sequence ($\{H, T\}$, 800 frames, around 15 s) from an individual user and output an adaptive feature ($z$) with a length of 1000. It was composed of 4 fully connected layers, each of which was followed with a ReLU activation function. The adaptive feature was then input into the forward model together with the haptic sequence. We trained individual forward models using the dataset from each participant, and a universal forward model without the adaptive module using the dataset from all participants as baselines. All models were trained with a learning rate of $1e^{-4}$, weight decay of $1e^{-4}$, and batch size of 32.

**Inverse haptics optimization.** Given a target tactile sequence ($T_{\text{target}}$), the inverse optimization outputs the optimized haptic instructions by interactively performing gradient descent over and minimizing the MSE between the predicted tactile sequence ($T$) and the target tactile sequence ($T_{\text{target}}$), which was output by the pre-trained forward dynamics model using the present haptic sequence as input (Fig. 4a). The initialization of the haptic sequence was extracted from the target tactile sequence by straightforward thresholding (the corresponding haptic units would be activated if the tactile signal exceeds a normalized threshold of 0.3). Gradient descent was performed with a learning rate of $1e^{-3}$, and weight decay of $1e^{-3}$.

**Validation.** Both offline and online evaluations of the pipeline were conducted. Qualitative results of the forward dynamics model are shown in Supplementary Fig. 9d. We first validated by comparing the optimized haptic instructions with the ground-truth pre-generated haptic sequence from the recorded dataset (Supplementary Fig. 9e). We further validated online by comparing the target tactile sequence with the user's output tactile sequence when given unoptimized and optimized haptic instructions (Supplementary Fig. 9f). More online validation was performed in piano playing, music rhythmic gaming[50], and car racing gaming[51] scenarios. Piano playing was performed on a digital piano (YAMAHA P-125) with a 5-finger control. The music rhythmic game was played on a laptop using a mobile game emulator (Mumuplayer), where the user directed the moving line by left-clicking the mouse with only their index finger. The car racing game was performed on a laptop through an online gaming platform, where users mostly controlled the balance and speed of the car by pressing direction keys using their middle and ring fingers. Validation and comparison of the effectiveness of our optimization pipeline were performed three times on two users. To eliminate bias, users were asked to get familiar with the validation scenarios before the experiments and to perform the tasks first with optimized haptic instructions, then with unoptimized haptic instructions, and lastly without any haptic instructions. Tactile information was recorded at 50 Hz and haptic instructions were generated at 100 Hz.

**Teleoperation**

In our teleoperation setup, users controlled a parallel gripper (Robotiq 2F-140) by moving their thumb and index finger. The distance between the two fingers was tracked by two embroidered color blocks, which were captured by the camera (Logitech C930e 1080p) and extracted by color thresholding using OpenCV[52]. The distance was converted to a command for the parallel gripper through linear mapping. Each of the parallel grippers was covered by a 3 by 6 pressure sensing matrix, which was averaged as 3 sensing read-out values. The participant wore a glove with 3 vibrotactile units along the thumb and 3 vibrotactile units along the index finger, which correspond to the individual sensing area on the parallel gripper. During the experiment, the contact interactions between the gripper and the object were captured by the tactile sensors in the form of 10-bit ADC output from the data acquisition circuit. We applied a threshold to the tactile signal at 25% of the full ADC output, meaning that any tactile signal exceeding this threshold was converted into haptic feedback and transferred to the user through the vibrotactile haptics. We asked the participants to adjust the distance between his/her thumb and index finger according to the haptic feedback they believed a stable grasp of the specific object was achieved. Such a procedure was performed when the participant was blinded and when the participant was offered a real-time view of the grasping process (Supplementary Movie 7). Tactile information was recorded at 50 Hz and haptic instructions were generated at 100 Hz with full PWM modulation.

## Data availability

The data that support the findings of this study have been included in the main text and Supplementary Information/Source Data file. The datasets for human behavior modeling and haptics optimization have been deposited at https://www.dropbox.com/home/adaptTacTransfer DropBox. All other relevant data supporting the findings of this study are available from the corresponding authors upon request. Source data are provided in this paper.

## Code availability

The codes to train and evaluate the human behavior modeling and haptics optimization pipeline are publicly available at https://github.com/yiyueluo/adaptTacTransfer GitHub along with the paper[46].

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

## Acknowledgements

We thank Yichen Li, Krishna M. Jatavallabhula, Wil Norton, Tiffany Louie, and Bolei Deng for their helpful suggestions on this work. This work is supported by the Schwarzman College of Computing Fellowship by Google and a GIST-MIT Research Collaboration grant. The authors are grateful for the support of Wistron, Toyota Research Institute, and Ericsson.

## Author contributions

Y. Luo implemented the hardware designs and performed characterization. Y. Luo, Y. Lee, and M.F. implemented driving circuits. Y. Luo and Y. Li conceived the adaptive human model learning and inverse haptics optimization pipeline. Y. Luo performed data collection and implemented the learning pipeline. Y. Luo and C.L. conceived and performed teleoperation demonstrations. Y. Luo and J.D. organized the results. K.W. implemented the magnetic flux simulation. D.R., T.P., Y. Li, A.T., and W.M. supervised the work. All authors contributed to the study concept, conceived of the experimental methods, discussed the results, and prepared the manuscript.

## Competing interests

The authors declare no competing interests.
