## [Peer Review File · Nature Communications]

REVIEWER COMMENTS

Reviewer #2 (Remarks to the Author):

The author developed a glove-like textile-based wearable human-machine interface with integrated tactile sensors and vibrotactile haptic actuators. This smart glove can record, reproduce, and transfer tactile interactions, showcasing the great potential for guiding people to perform physical skills and enabling responsive robot teleoperation. However, more detailed descriptions and additional evidence should be provided to support the author's claim. Therefore, this manuscript cannot be recommended to be accepted at the current stage. I would reconsider the manuscript depending on the authors' responses. The followings are some detailed comments:

1. Although non-woven fabrics are used as substrates for smart gloves, integrated Magnets and copper wires inevitably increase the weight of the glove and produce a foreign body sensation, probably weakening the wearable comfort.
2. As a wearable human-machine interface, great mechanical stability is also required for the glove, besides the breathability. Whether can the glove work well after being twisted, warped and folded?
3. The tactile sensor is used to sense the applied force and produce corresponding electric signals as the input information, which plays an important role in this interface. Could the author provide more experiment data about the sensing performance of the used tactile sensors, such as the sensitivity, the minimum detection limit, and response/recovery time and, so on
4. As an interactive system with integrated tactile sensors and vibrotactile haptic actuators, its delay time has an enormous effect on the user experience. Therefore, could the author provide more details about the delay time of each stage, including the sensing of tactile sensors, the calculating of models, and receiving and executing of actuators?
5. Could the author give some details about the control circuit of this interactive system to more clearly describe the complete interactive process?

Reviewer #3 (Remarks to the Author):

The author presented a textile-based human-machine interface with integrated tactile sensors and vibrotactile haptic actuators. And the experiments demonstrate several applications with teleoperation.

This article truly has a variety of demos, but lack of clear experiment data and optimization processes. It is hard to accept this manuscript in this journal as its form now.

1. As an important component, resistive tactile sensors do not have enough characterizations, even Fig 2g did exhibit the multiple force with certain force (5 N). How about the sensitivity, the linear range, or the reproducibility? Whether the tactile sensors can give more details on the feedback?
2. With the highlighted digital embroidery machine, the smart gloves were produced. By different sizes of subjects' hands, how can the machine adjust and fabricate the gloves rapidly? Maybe more efforts can be put on this concept.
3. It is unclear with the number of samples for the confusion matrices in the human perception experiments. 10 subjects size was mentioned, but how many cycles and what was sample size?
4. In Fig. 5b, the Y axis units were missing, and neither the Fig. 4. It is necessary to detail what the numbers mean.

We would like to thank the reviewers for their invaluable comments and suggestions. We have carefully considered all the reviewers' comments and revised the manuscript accordingly. A point-by-point response letter to reviewers' comments is included. All responses and changes are marked in blue. Thank you for your consideration and look forward to your response.

REVIEWER COMMENTS

Reviewer #2 (Remarks to the Author):

The author developed a glove-like textile-based wearable human-machine interface with integrated tactile sensors and vibrotactile haptic actuators. This smart glove can record, reproduce, and transfer tactile interactions, showcasing the great potential for guiding people to perform physical skills and enabling responsive robot teleoperation. However, more detailed descriptions and additional evidence should be provided to support the author's claim. Therefore, this manuscript cannot be recommended to be accepted at the current stage. I would reconsider the manuscript depending on the authors' responses. The following are some detailed comments:

1. Although non-woven fabrics are used as substrates for smart gloves, integrated Magnets and copper wires inevitably increase the weight of the glove and produce a foreign body sensation, probably weakening the wearable comfort.

Authors: Thank you for sharing your insightful comments and concerns on the integration of copper wires and magnets. We included further discussion in the revised manuscript, Discussion section, and Supplementary notes. Our integrated permanent magnets are comparable in size to rigid buttons, fasteners, and zippers commonly found on garments (6.35 mm in radius and 1.6 mm in thickness). Therefore, we consider our system a textile-based wearable interface. Each permanent magnet weighs 0.377 g and the enameled copper wire weighs less than 1 g per meter. In our full-sized smart glove designs, the 23 vibrotactile haptic units and connecting traces add less than 15 g to the original glove. For reference, the plain spandex-based glove weighs around 10 g, and the plain commercial thick handling glove for the tactile interaction transfer demonstration weighs 200 g. As part of our future work, we suggest the substitution of permanent magnets with soft magnets, aiming to create a fully soft system, although this may result in weaker magnetic fields.

2. As a wearable human-machine interface, great mechanical stability is also required for the glove, besides the breathability. Whether can the glove work well after being twisted, warped and folded?

Authors: We appreciate your feedback on the mechanical properties of our smart gloves. We performed characterization on tensile and bending tests and included results in the revised manuscript (Result section, Supplementary note, and Supplementary Fig. 4).

We investigated the durability of non-woven fabric embroidered with enameled copper wire by subjecting a 20 cm sample to 3000 bending cycles. As demonstrated in the following figure (a), the bending cycles were performed in a tensile test setup, where the sample was placed at the initial positions with a minimum bending radius of 3.18 cm. It was then stretched with a tensile length of 10 cm. The following figure (b) shows photos of the embroidered vibrotactile coils at 0, 1000, 2000, and 3000 cycles. Notably, no significant changes were observed. Throughout our user studies and experiments, our gloves have endured over 6 hours of use, involving more than 100 instances of wearing and removal, with no noticeable performance degradation in the tactile sensors and vibrotactile haptic units.

The glove is built on commercial fabric (non-woven fabric and Spandex fabric). We conducted tensile tests on pure non-woven fabric, pure Spandex fabric, and fabric that incorporated embroidered enameled copper wire. The tests were conducted on 20 cm samples using a Shimadzu AGS-X mechanical tester, operating at a speed of 20 mm/min. The considerations of mechanical properties have been included in the Result section. The stress-strain curves are illustrated in Figure (a) below.

The pure non-woven fabric substrate exhibits a tensile strength of approximately 28 N, with a maximum elongation of 75%. When non-woven fabric is embroidered with enameled copper wire, the sample maintains a similar Young's modulus but breaks at a maximum elongation of 15%, primarily due to the enameled copper wire's fracture, as depicted in Figure (b). On the other hand, the pure Spandex fabric enables maximum elongation of up to 180%. Spandex with embroidered copper wires breaks at a maximum elongation of 25%, with greater elasticity and lower Young's modulus (as evident in the light blue curve in the plot). We speculate that the increased elongation is attributable to the smoother surface of Spandex fabric, resulting in reduced friction and tolerating greater movement of the enameled copper wire during stretching. A glove typically endures elongation in the range of

10%-20% during daily activities; therefore, it is reasonable to assume that our smart gloves can withstand stretching during the bending and movement of our hands.

3. The tactile sensor is used to sense the applied force and produce corresponding electric signals as the input information, which plays an important role in this interface. Could the author provide more experiment data about the sensing performance of the used tactile sensors, such as the sensitivity, the minimum detection limit, and response/recovery time and, so on

Authors: We appreciate your feedback on the performance and consistency of the sensors. We conducted additional characterization on our tactile sensor and updated our results in the Result section, Fig. 2 and Supplementary Fig. 3.

As demonstrated in the following figure, our typical tactile sensor obtains a minimum detection limit of 0.35 N/cm^2 and a maximal detection limit of 20 N/cm^2 . The linear region I and II obtain a sensitivity of $\sim 1000 \Omega/(\text{N/cm}^2)$ and $\sim 25 \Omega/(\text{N/cm}^2)$ respectively. It is worth noting that the embroidered conductive traces are affixed in close proximity but not in direct contact with the piezoresistive film by the surrounding adhesive. Consequently, an initial force is necessary to establish contact, serving as the primary limiting factor for the sensitivity of our tactile sensor.

We conducted additional characterizations to assess the response and recovery time of the sensor. These tests involved loading and unloading cycles at different speeds, indicated by strain. Each complete loading-unloading cycle ranged from 2 seconds (10 mm/min) to 20 seconds (1 mm/min). The tactile sensor exhibited similar and relatively small hysteresis under varying loading-unloading speeds. Given our specific usage scenario, where users engage in low-frequency pressing, we do not anticipate significant high-frequency signal changes.

4. As an interactive system with integrated tactile sensors and vibrotactile haptic actuators, its delay time has an enormous effect on the user experience. Therefore, could the author provide more details about the delay time of each stage, including the sensing of tactile sensors, the calculating of models, and receiving and executing of actuators?

Authors: Thank you for your comments. We included additional information in the Supplementary Notes. In general, we consider that the tactile sensors transmit signal and the vibrotactile haptic actuators output response with negligible time delay. More specifically, the tactile sensing data was serialized to a laptop at the frame rate of 60 Hz, with a time delay of 16 ms. The model takes less than 1 ms to output the optimized haptic sequence with the use of NVIDIA RTX 3080. The vibrotactile haptic actuators are controlled by Arduino Mega 2560 Rev3 with a clock speed of up to 16 MHz. The microcontroller receives and executes actuating signals at each digital pin with a baud rate of 115200 bits per second. The execution of the vibrotactile actuators is controlled by an array of H-bridges consisting of N-MOSFETs (IRLB8721PbF, Infineon Technologies). According to the spreadsheet and as demonstrated in the following figure, during switching, the turn-on delay time ($t_{d(on)}$) is 9 ns, the turn-off delay time ($t_{d(off)}$) is 9 ns, and the rise time (t_r) and fall time (t_f) are 91 ns and 17 ns respectively.

For the teleoperation application specifically, the delay also comes from the capturing of users' finger position via webcam and the estimation of the distance between fingers, which is around 20 Hz, with a time delay of 50 ms, as well as the transmission of distance information to the robot gripper controller, which is running at 50 Hz.

Overall, the time delay from the capturing of tactile signal, and the optimization, transmission and execution of the vibrotactile haptic sequences is less than 30 ms. There will be a perception delay, i.e., the time lag between the haptic signal output and the user's actual actions. Such delay is inevitable and our goal is to optimize the haptic output to minimize the effect of such delay on the users' performance.

5. Could the author give some details about the control circuit of this interactive system to more clearly describe the complete interactive process?

Authors: We appreciate your feedback on the control circuit of the interactive system. Below is an overview of our tactile interaction transfer process. We first extract the tactile interaction from the expert's action via the tactile sensor acquisition circuit. This tactile data is then fed into the adaptive learning and inverse optimization pipeline, which generates the optimized haptic sequence. The optimized haptic sequence is subsequently received and executed by the vibrotactile haptic driving circuit to provide real-time feedback for guiding the novice's actions. We extended our description of our vibrotactile haptics driving circuit in Method and included the transfer pipeline in Supplementary Fig. 9.

We updated the description of the driving circuit in the Method section (Section 4.7 paragraph 2) as the following. The vibrotactile haptic units in our system were controlled through a customized circuit integrated into a microcontroller board (Arduino Mega 2560 Rev3). Our circuit is composed of several key components, including a matrix of five H-bridges, Schottky diodes, and connectors for interconnecting the vibrotactile haptic units. The H-bridge matrix served as the central component for directing current to the haptic units. Within this matrix, each row and column are connected to a half-bridge circuit architecture, comprised of two N-MOSFETs (IRL8721PbF, Infineon Technologies) connected in series. The vibrotactile haptic units were activated in a multiplexed manner with this H-bridge matrix and a microcontroller board. The microcontroller board first selects pairs of horizontal (e.g. A11 and A12) and vertical (e.g. A12 and A22) half bridges to form a full H-bridge configuration, utilizing digital pins (e.g. A12 and A22). The microcontroller board then drives the vibrotactile haptic units with controlled intensity and timing through pulse-width modulation (PWM) pins (e.g. A11, and A21). The microcontroller was programmed and constantly updated with an optimized haptic sequence generated through an adaptive learning and inverse optimization pipeline. Each vibrotactile haptic unit was isolated by a Schottky diode (1N4001) to prevent interference and cross-talk between the haptic units. External power was supplied by a portable lithium-ion polymer battery (3.7 V, 2500 mAh) to deliver the necessary power to coils to drive the vibrotactile haptic units.”

Reviewer #3 (Remarks to the Author):

The author presented a textile-based human-machine interface with integrated tactile sensors and vibrotactile haptic actuators. And the experiments demonstrate several applications with teleoperation. This article truly has a variety of demos, but lack of clear experiment data and optimization processes. It is hard to accept this manuscript in this journal as its form now.

1. As an important component, resistive tactile sensors do not have enough characterizations, even Fig 2g did exhibit the multiple fore with certain force (5 N). How about the sensitivity, the linear range, or the reproducibility? Whether the tactile sensors can give more details on the feedback?

Authors: Thank you for your feedback on the sensor performance. We performed additional characterization and results are updated accordingly in the revised manuscript (Fig. 2g and Supplementary Fig. 3). Our typical tactile sensor obtains a sensitivity of 0.35 N/cm^2 , with a maximal detection limit of 20 N/cm^2 , as addressed in comment 3 from Reviewer #2. We performed additional characterization on the sensing performance of 3 individual tactile sensors with 5 loading-unloading cycles. As demonstrated in the below figure, individual sensor obtains similar sensing performance, with 2 linear regions. To better quantify the pressure, we switch the unit of force (N) into the unit of pressure (N/cm^2).

We further demonstrated the stability and durability of our sensor by subjecting three individual tactile sensors to a 2000-cycle test. The results are presented in the following figure.

2. With the highlighted digital embroidery machine, the smart gloves were produced. By different sizes of subjects` hands, how can the machine adjust and fabricate the gloves rapidly? Maybe more efforts can be put on this concept.

Authors: We appreciate your comments and suggestions on the design and fabrication pipeline of the smart gloves. We have streamlined the rapid and customized design and fabrication of our gloves. We start by capturing an image of the user's hand with a fixed reference point. From these images, we extract crucial hand size parameters, including finger length, palm width, and more. These parameters can then be applied to a predefined glove design template within any vector-based interactive user interface, such as Adobe Illustrator or Inkscape. While we maintain the predetermined arrangement of tactile sensors and vibrotactile haptics, it's important to note that their placement and design parameters can be effortlessly tailored to suit individual user requirements. The customized glove designs are then converted into embroidery machine files with defined pitches via a Python script and automatically fabricated using the machine. The pipeline is illustrated in the below figure and has been added to Supplementary Fig. 4a. In future work, we would like to develop a more complex computation design pipeline that incorporates an interactive user interface and automatic parameter fine-tuning.

Below are photographs of fabricated gloves in small (12 cm x 23 cm), medium (11 cm x 20 cm), and large sizes (10 cm x 17 cm). It is noted that the embroidered vibrotactile coils obtain the same size and positions while the glove outline shrinks and expands according to the hand size. The result has been added to Supplementary Fig. 4b.

3. It is unclear with the number of samples for the confusion matrices in the human perception experiments. 10 subjects size was mentioned, but how many cycles and what was sample size?

Authors: Thank you for the question. We clarified the results from the user study in the caption of Fig. 3 and the Method section. For the position identification study, we actuated a random vibrotactile unit and asked the participant to identify the location 30 times. All vibrotactile units were covered but not with the same number of cycles. For the temporal sequence identification study, we randomly played one of the 6 vibrotactile sequences and asked the participant to identify it. This process repeats for 10 times.

4. In Fig. 5b, the Y axis units were missing, and neither the Fig. 4. It is necessary to detail what the numbers mean.

Authors: Thank you for raising the concern on clarity. We converted the distance between grippers into physical units in Fig. 5b. For the tactile signal, however, we only captured the 10-bit ADC output from the data acquisition circuit during our teleoperation experiments. We applied a threshold to the tactile signal at 25% of the full ADC output, i.e., any tactile signal exceeding this threshold was converted into haptic feedback during the haptics-enabled scenarios.

REVIEWERS' COMMENTS

Reviewer #2 (Remarks to the Author):

After the revision, I think this manuscript can be accepted now.

Reviewer #3 (Remarks to the Author):

The comments are addressed, I recommend the manuscript published here.